# Selected Healthy Behaviors and Quality of Life in People Who Practice Combat Sports and Martial Arts

**DOI:** 10.3390/ijerph16050875

**Published:** 2019-03-10

**Authors:** Katarzyna Kotarska, Leonard Nowak, Mirosława Szark-Eckardt, Maria Nowak

**Affiliations:** 1Department of Physical Culture and Health Promotion, University of Szczecin, 71-004 Szczecin, Poland; leonard.nowak@usz.edu.pl (L.N.); maria.nowak@usz.edu.pl (M.N.); 2Faculty of Physical Education, Health and Tourism, Kazimierz Wielki University in Bydgoszcz, 85-064 Bydgoszcz, Poland; szark@ukw.edu.pl

**Keywords:** healthy behaviors, quality of life, combat sports and martial arts, WHOQOL-BREF questionnaire

## Abstract

*Background*: The quality of life of a society is conditioned by many factors, and depends, among other things, on preferred behavior patterns. Combat sports (CS) and martial arts (MA) have a special educational potential in the area of shaping positive behavior patterns and transmitting moral values which could help reduce aggression in society. The aim of the work was to determine the relationship between health behaviors and the quality of life of people who practice combat sports and martial arts (CS and MA) recreationally, in addition to practicing other sports, and as competitors at the master level. *Methods*: The research embraced 543 people who practice combat sports and martial arts. Three groups were selected: recreational (*n* = 362), people who reconciled practicing various sports (*n* = 115), and competitors who practiced combat sports or martial arts at the master level (*n* = 66). The average age of the respondents was 24.49 ± 7.82. The standardized WHOQOL-BREF questionnaire and another questionnaire for a lifestyle survey were applied. The Kruskal-Wallis test was used to compare several independent samples. In the case of determining statistical significance of differences the Mann-Whitney test was employed, and for the qualitative data analyses the trait frequency and the independence chi-square test were used. The effect size was calculated for each test ($E_{R}^{2}$, rg, Cramér’s V). The value of *p* ≤ 0.05 was assumed to be statistically significant. *Results*: The highest quality of life (in the physical, psychological and environmental domains) was characteristic of the competitors, who practiced only combat sports and martial arts. They also displayed the most health-oriented behaviors. The surprising results were: lower quality of life in the assessment of nondrinkers and nonsmokers, and higher among people who were overweight. *Conclusions*: We have found positive correlations between practicing CS and MA, health behaviours and higher scores in quality of life self-evaluation, particularly where practitioners are exclusively focused on CS and MA and practice these at a competitive level. Our findings thus support the growing evidence that competitive level CS and MA are an effective means of improving people’s quality of life. Future research needs to clarify whether CS and MA can also be recommended to recreational and non-competitive practitioners as a means to improve their subjective quality of life.

## 1. Introduction

In the light of some research, combat sports (CS) and martial arts (MS) are popular and rapidly developing sports disciplines [1], having a large group of supporters who seek varied sensations and emotions. Analyses based on a representative sample of Polish people, concerning, among others, the frequency of practicing combat sports and martial arts (CS and MA), do not indicate society’s particular interest in this kind of sporting activity. Out of 12,405 respondents, 470 (3.8%) declared having basic skills in combat sports or martial arts, while only 124 (1%) reported practicing them [2]. 

Many publications appreciate the rich potential of this sport, which can be used in various areas of life, including the shaping of psychophysical well-being, harmony of the body and spirit, reducing aggression [3,4], increasing one’s own safety. Regular training positively influences their well-being, personality development, and cognitive and educational functions [5,6]. Multidimensional benefits, such as improvement of psychophysical well-being, increased self-defense possibilities, and avoidance of addictions, were observed among teenagers practicing taekwondo [7]. 

At the same time, there are many common, controversial opinions about the effects of practicing CS and MA. The abovementioned possibilities of shaping positive personality traits, and the occurrence of aggressive and antisocial behavior, have not yet been settled definitively [8]. Many researchers, however, have shown associations between the inclination to choose health-oriented behaviors and the quality of life of people who practice sports at different levels [9,10]. 

Physical, psychological, social and cognitive functioning and general well-being compose the five dimensions of the quality of life conditioned by the state of health. The quality of life is connected with the general sense of health understood as “... a state of complete physical, mental and social well-being” [11]. Quality of life surveys stress the necessity to consider factors such as gender, age, education, family status, interests, and, particularly, occupational status and net income per household [12]. All these elements influence health-oriented choices and, consequently, the quality of human life.

The perception of the quality of life is subjective, dependent not only on one’s state of physical and mental health, self-reliance, social relations or environmental factors, but also on their life position, interests, life activity, attitudes and adopted systems of values. Many studies into the quality of life concern people who are ill [13], but there are also studies into the quality of life of the healthy populations in the context of lifestyle, forms of spending free time, and physical exercise [14]. The relationships between the increase in the level of physical activity and the quality of life are not conclusive. Higher quality of life was found in people who engaged in moderate physical activity in their free time than in those who exercised intensively. The quality of life of intensive exercisers was comparable to that of physically passive people who did not engage in physical activity in their free time [15,16]. Comparative studies into the quality of life of professional athletes and amateurs indicate higher quality of life in individuals who are more exposed to severe competition. It was confirmed that competition-based sporting activity promotes health and helps in the pursuit of a healthy lifestyle [17]. The quality of life depends on the level of competition, but also on gender, higher quality being men’s characteristic [18]. In the face of increasing effects of civilization diseases, it is important to create preventive programs focused on replacing risky behaviors with healthy ones [19,20,21] This is especially important for athletes, since health is seen as a prerequisite to achieving sporting success and should also be a result of doing sports. No studies were found in literature which would concern the comparison and analysis of health behaviors in people who practice CS and MA recreationally, engage in them in addition to practicing other sports, and have remarkable achievements in them, in the context of their quality of life. So far, studies most often focused on a particular combat sport or martial art [7,22,23,24]. 

The aim of the work was to determine the relationship between health behaviors and the quality of life of people who practice combat sports and martial arts recreationally, in addition to practicing other sports, and as competitors at the master level.
People who practice combat sports and martial arts at a competitive level display a higher level of healthy behaviors than individuals who practice these sports recreationally or reconcile practicing various sports.
People who practice combat sports and martial arts at a competitive level have higher quality of life than individuals who practice these sports at non-competitive levels. Higher quality of life is characteristic of subjects who display health-oriented behaviors.

## 2. Material and Methods

The study included 543 individuals who practiced CS and MA in 18 clubs in the following Polish provinces: Kuyavian-Pomeranian, Lubusz, and West Pomeranian. The respondents practiced: boxing, Brazilian jiu-jitsu, karate, Mixed Martial Arts, and muaythai (Thai boxing). Three groups were selected: 362 people who considered practices recreation (G I), 115 athletes dedicated to other sports (e.g., soccer players, track and fielders, canoeists), who complemented their general preparation through participation in combat sports or martial arts practices (G II), and 66 combat sports and martial arts competitors with significant sports achievements at the level of at least the Polish Championships (G III).

The average age of the respondents was 24.49 ± 7.82. Men comprised 54.7% of the subjects. The subjects were mainly city dwellers (88.1%). Single persons accounted for 72.1%, married for 21.4%, and 6.5% cohabitated. 39.4% of the respondents were still at school or university, 14.7% reconciled study and work, 24.8% performed mental work, 14.9% physical work, and 1.5% ran their own businesses. Unemployed persons accounted for around 5% of the subjects.

The diagnostic survey method was employed and the standardized World Health Organization Quality of Life—BREF (WHOQOL-BREF) questionnaire was used to assess the quality of life. A shortened version, adapted to Polish conditions in terms of language, culture and psychometrics [25] comprised 26 questions. The first two questions regarding the level of satisfaction with life and health were analyzed separately. The remaining 24 questions concerned the four domains of the quality of life. In the physical domain the assessment included: activities of daily living, dependence on medicinal substances and treatment, energy and fatigue, mobility, pain and discomfort, rest and sleep, and work capacity. In the psychological domain assessed were: enjoyability and meaningfulness of life, ability to concentrate, bodily appearance, self-esteem, and moods. In the social domain the subjects were to assess: personal relationships, sexual activity, and social support. The environmental domain concerned the assessment of: safety, physical environment, financial resources, opportunities for acquiring new information and skills, participation in recreation and leisure, home environment, health care accessibility, and transport. The WHOQOL-BREF facilitates characterization of the quality of life profile within the four domains mentioned above. The score for each domain is determined by calculating the arithmetic mean for the items included in a particular domain. The greater the number of points in the domains, the higher the quality of life. The maximum value is 120 points [25].

In the research, the original survey technique for examining lifestyles of people who practice sports was also employed. In this work, data concerning age, gender, place of residence, education, type of work, and financial situation were used. 

The analysis covered selected behaviors which positively or negatively influence health: practicing CS and MA (length of training history, frequency and weekly time devoted to exercise), pattern of eating measured by the BMI index, having dental checkups, consumption of alcoholic beverages, and smoking. The focus was on establishing the relationships between these behaviors and practicing sports by representatives of particular groups and the general assessment of the quality of life. 

Written informed consent was obtained from each subject included in the study. The study protocol was approved by the appropriate Ethics Committee of Kazimierz Wielki University No. KEBN 7/2018 and conformed to the ethical guidelines of the 1975 Declaration of Helsinki. The overall results obtained in the WHOQOL-BREF were divided according to the quartiles and median into four groups. A similar procedure was applied with respect to the length of practicing sports by the respondents. Nonparametric statistics were applied in the analyses of the results (distribution that differs from normal). The Kruskal-Wallis test (H) was used to compare several independent samples. In the case of deter) mining statistical significance of differences for the comparison of two independent samples, the Mann-Whitney (U) test was employed. In qualitative analyses, the trait frequency and the independence chi-square test were used. The effect size was calculated for each test: E^2^_R_ for the Kruskal-Wallis H test, Glass rank biserial correlation (rg) for the Mann-Whitney U test, Cramér’s V for the χ^2^ test. The value of *p* ≤ 0.05 was assumed to be statistically significant. Statistical calculations were made with the Statistica for Window 12 software (StatSoft Sp. z o.o., Crakow, Poland, and Microsoft Office Excel 2007 (Microsoft Sp. z o.o., Warsaw, Poland). 

## 3. Results

Individuals who practiced combat sports and martial arts differed statistically significantly with regard to age (*p* = 0.0220 for the χ^2^ test) (Table 1). Persons who treated practices as a form of physical recreation (G I) were in all age categories. Those who practiced other sports, complementing their general preparation through combat sports or martial arts training (G II) were mostly under 24 years of age (58.3%). In the group of competitors (G III) there were people who were under 19 as well as aged 24–28 (36.4% and 31.8% resp.). The respondents most often had a higher or secondary education. The greatest number of people with a higher education were in G I and G III, and with a secondary education in G II. Among the competitors, in G III, the proportion of individuals who were still during school education was higher (28.1%) (*p* = 0.0008 for the χ^2^ test). The respondents most often had a good financial situation (48.3%). Very good and good ratings were characteristic of people from G III (39.4%, 53%). At the same time, about 20% of the respondents from G I and G II assessed their financial situations as fair (*p* = 0.0185 for the χ^2^ test). The respondents were characterized by varied lengths of practicing combat sports and martial arts (*p* = 0.0151 for the χ^2^ test). In G I the most people had practiced the sport for less than four years (63.3% in total). Over 53% of the subjects in G II and G III had participated in combat sports or martial arts training for over four years. The greatest number of respondents devoted from 181 to 360 minutes a week to training (38.5%). They were most often G I and G II subjects (42.5% and 32.2% resp.). The competitors (G III) were among those who exercised over 361 minutes a week, and their number grew in successively increasing time ranges (*p* = 0.0000 for the χ^2^ test). Among those exercising over 720 minutes a week (12 hours or more a week) there were 25.7% of G III subjects. The weekly exercise time was associated with the weekly training frequency. The competitors exercised with the greatest frequency, twice or once a day (13.6% and 42.4% resp.) (*p* = 0.0000 for the χ^2^ test). G I and G II were characterized by lower exercise frequency, 3–4 times a week or less often.

The respondents reported various healthy behaviors (Table 2). The competitors (G III) avoided smoking more often than the subjects from G I and G II (*p* = 0.0224 for the χ^2^ test), and abstained from alcohol (*p* = 0.0174 for the χ^2^ test). The athletes who practiced different disciplines (G II) more often declared consumption of low-alcohol beverages (46,5%). 17.6% of the respondents consumed only high-alcohol beverages, the competitors (G III) being less likely to so (13.6%). Most of the subjects had a normal BMI (54.9%). Overweight (28.2%) and obese (4.6%) people were also observed. Differences between the groups were not statistically significant. 54.5% of the respondents had dental checkups (every six months, in accordance with the preventive recommendations), and they were more often people from groups G II and G III (59.7% and 66.8% resp.) (*p* = 0.0012 for the χ^2^ test). In the past year, 29.6% of the subjects in G I had had the checkups. There was also a small group of individuals (3%), who had not had a dental appointment for over two years. The measures employed show a small effect size.

Satisfaction with life, health and particular domains of the quality of life of people who practice combat sports and martial arts were presented in Table 3. No differences were found with regard to life satisfaction. The people in G II were less satisfied with their health than those in G III (*p* = 0.0403). The quality of life in the physical domain was rated higher by G I subjects than by G II ones (*p* = 0.0008); G III also evaluated it higher than G II (*p* = 0.0046). In the psychological domain of the quality of life differences between G III and G II were observed (*p* = 0.0074): in G III the assessments were higher. The subjects in G II evaluated their quality of life in the social domain higher than those in G I (*p* = 0.0487). As far as the quality of life in the environmental domain is concerned, all groups differed from each other. G III assessed the components of this domain higher than G I (*p* = 0.0238), who in turn had a higher evaluation compared with G II (*p* = 0.0092). A significantly lower assessment of the quality of life was found in people in G II in comparison with those in G III (*p* = 0.0001). A large effect size was observed in different domains of the quality of life (satisfaction with health, and the physical and psychological domains) between G II and G III. A smaller effect size occurred with respect to G II and G III in the environmental domain.

The diversity of satisfaction with life, health and different domains of the quality of life for selected healthy behaviors of the subjects were presented in The nonsmokers were more satisfied with their health in comparison with the smokers (*p* = 0.0155), who assessed their quality of life in the social domain higher (*p* = 0.0480) (Table 4). Statistically significant differences were found in the social domain among the people who consumed various kinds of alcoholic beverages. The lowest quality of life was reported by the nondrinkers as compared with those who consumed low-alcohol (*p* = 0.0000), high-alcohol (*p* = 0.0000) (Table 5). 

The quality of life in the physical domain was rated higher by the subjects who had had dental checkups in the past six months than by those who had had their teeth examined over a year but less than two years before (*p* = 0.0019) (Table 6). A similar relationship was observed between dental checkups in the period 6–12 months before and 1–2 years before (*p* = 0.0142). Those who had had dental examinations 6–12 months before, reported lower quality of life in the social domain (*p* = 0.0070). In the environmental domain, lower assessments were characteristic of the subjects who had had dental checkups 6–12 months before (*p* = 0.0078) or 1–2 years before, as compared with those who had undergone dental examinations in the past 6 months (*p* = 0.0231). 

In the social domain, the large effect size indicates differences between the nondrinkers and those who consumed high-alcohol and various beverages. In the physical, social and environmental domains, the large effect size indicates differences between the subjects who had had dental check-ups in the past 6–12 months or 1–2 years and those who had visited the dentist earlier.

## 4. Discussion

Combat sports and martial arts affect comprehensively the personality development of those who practice them: they lead them to a high level of physical fitness, build their mental resistance and courage, and help them to eliminate negative emotions. Athletes’ health-oriented lifestyle—as a recommendation in maintaining health and high quality of life—should be characterized by consistency and maintenance of a high level of behaviors which lead to, among others, outstanding sports results. Choosing health-oriented behaviors contributes to maintaining health and high quality of life. Athletes’ anti-health behaviors block the path of development in their sporting careers. A study of a large group of teenagers (2659) aged 14–16 proved that athletes achieve higher results in terms of overall quality of life and in all its spheres compared with healthy individuals who do not practice sports [10]. 

In this study, the hypothesis was confirmed that CS and MA competitors (G III) are characterized by a higher level of healthy behaviors in comparison with people who practice these sports recreationally or reconcile practicing various sports. The competitors were the least likely to smoke, half of them were nondrinkers, and most of them had dental checkups. 

Undoubtedly, this is connected with the choice of a sports discipline, because athletes who achieve great results in CS and MA prefer health-oriented lifestyles and patterns of health-related behaviors. Such sports disciplines require willpower, restraint, and perseverance in action [26]. 

Slightly different behaviors were preferred by the subjects who practiced combat sports recreationally (G I) and the athletes from other sports disciplines who additionally practiced combat sports (G II). The latter ones more frequently smoked and consumed all kinds of alcoholic beverages. They had dental checkups in accordance with the preventive recommendations. Dental health-oriented behaviors, which also include the optimal frequency of dental examinations, are not fully implemented in Polish society. A study of 18-year-old high school students conducted in Lodz schools showed that only 40.7% of them had had dental checkups in the past 6 months [27]. The respondents who practiced CS and MA had more often undergone such checkups (54.5% in total). But this level of health-oriented dental behaviors of young people is not satisfactory.

Athletes are required to undergo systematic medical examinations, including dental ones, which is why such behaviors were expected. The persons who engaged in CS and MA exclusively for recreation (G I) consumed mostly various kinds of alcoholic beverages and were less likely to have prophylactic dental examinations. Practicing sports at a competition level—and this would confirm our research results—is a protective factor with regard to smoking, but not to the consumption of alcoholic beverages. The slightly higher consumption of alcoholic beverages found in groups of soccer players seems to be comparable to that observed among handballers [28]. In a study of 101 judoka, 24 of whom had master ranks, a high level of healthy behaviors was found in 51.5% of men and 42.4% of women. It was shown that the average values of the general index of healthy behaviors were higher than the value obtained by adult Poles’ populations. It was also found that the level of those behaviors decreased with age, length of training, and improvement in athletic performance [22,29]. 

It can be concluded that long-lasting, tedious training experience and being repeatedly in the same sports environment (camps) can reduce the indexes of health-related behaviors. Moreover, practicing sports imposes on athletes permanent daily training regimes. Long-term sporty lifestyles can be burdensome for them and may result in a desire to react in an unhealthy way [19].

The assumptions of CS and MA can be a source of inspiration for changes in athletes’ way of thinking. Davis and Menard [20] showed that in contact sports, especially martial arts, the health dimension is higher, which stems from the fact that they constitute not only a defense method, but primarily a body strengthening method and biological regeneration, as well as prophylaxis and therapeutic action. Athletes reported a change in the approach to everyday problems as well as a change in the lifestyle as a result of practicing martial arts. The hypothesis was also confirmed in the research that the respondents were characterized by different assessments of the quality of life. It was shown that the highest quality of life is characteristic for competitors from group III, training sports and martial arts, who have significant sports achievements at the level of the Polish Championships. A large effect size was observed in different domains of the quality of life (satisfaction with health, and the physical and psychological domains) between G II and G III. A smaller effect size occurred with respect to G II and G III in the environmental domain. Only in the social domain the athletes from other sports disciplines dominated. In all three groups, the subjects perceived that the quality of their lives was at a relatively high level, and the differences concerned only particular dimensions. It may be connected with the specific type of discipline. This is confirmed by a study of former professional table tennis players, in which high levels of the quality of life and satisfaction with life were observed [9]. Team sports, in which the mental burden is distributed among different players, are preferred by emotionally labile individuals, because group action involves fewer of their resources and is safer than individual action. To act in a particularly difficult situation, the team or coach may appoint a person with greater mental resistance. In individual sports, the athlete bears a huge load, both physically and mentally, without the support of teammates.

Research presented by Argyle [30] indicates that sport is a reliable way of positively affecting the quality of life. Incorporating physical exercise programs in the treatment of depression or anxiety disorders increases the subjective sense of happiness. Sport and physical activity have a significant impact on the perception of the quality of life [31], not only because of the release of endorphins, but also because of interactions with other people, which is particularly satisfying for extroverts. 

In a study of teenagers, a positive influence of practicing CS and MA on their well-being was shown. Comprehensive elements of the training help young people to meet different needs, expectations and requirements, and as a result improve their well-being [7]. Research results from existing scientific literature indicate that martial arts (karate, taekwondo, kung fu and judo) can improve some aspects of cognitive functions and neurotrophic factors (BDNF and IGF-1) associated with the work of the brain. In particular, martial arts can stimulate the development of cognitive functions in children and adolescents and slow down the decline in cognitive ability of adults in middle and old age [6].

In a study (questionnaire and Internet) of 343 athletes who practiced CS and MA, the subjects evaluated their own health and overall quality of life higher than the general public. Of the parameters assessed, greater differences were observed in physical aspects of health than in psychological ones. The influence of practicing this kind of sport on health, the quality of life and the choice of health benefiting behaviors was confirmed [23]. These results mostly confirm the outcomes of the present study concerning the group of competitors (G III).

An English study into the quality of life (HRQoL) which embraced 5537 adults is also interesting. Both subjective (questionnaire) and objective (accelerometer-aktigraf model GT1M) measurements of physical activity were used. The use of an objective measure of physical activity shows a relatively better quality of life [32]. The use of questionnaire surveys seems therefore relevant.

The hypothesis that health-oriented behaviors of the respondents enhance their quality of life was also confirmed. Through an analysis of the correlations between those behaviors and the quality of life, it was noted that in the social domain lower quality of life appeared in the assessments of the nondrinkers and the nonsmokers, and higher quality of life was reported by the overweight subjects. The social domain includes, among others, the evaluation of personal relationships and social support. The social group has a profound impact on an individual’s conduct, especially a young person. Young people (more than a half under 23 years of age) who consistently adhere to the principles of a healthy lifestyle can be isolated in their social group if they shun smoking and drinking alcoholic beverages. The issues of cigarette smoking, alcoholism and drug addiction have not lost interest among young people. An athlete who wants to achieve the highest sports results has to choose patterns of behaviors which enhance health—and consequently their athletic performance—and put them first.

Combat sports and martial arts competitions consist in direct, contact fights between the contending athletes, and the weight limits in each weight category are the formal conditions for admission to the competition. Depending on the discipline category, they are often lower or higher than the normal BMI.

In the present study, it was shown that the majority of subjects had a normal BMI, which is logical and confirmed. Competitors are required to adjust their weight to a given weight category at a given time; monitoring the maintenance of appropriate body weight is common. Through a proper dosage of physical exertion during practices and proper nutrition they control their body weight [24]. Practicing martial arts for 12 weeks (3 hours a week) is also a feasible exercise intervention in adult overweight and obese women and can have a positive effect on weight control and overall health. In addition, engaging in this kind of activity significantly improves the quality of life in the physical, emotional and mental aspects, also in the population of overweight and obese women [33]. In review studies, results concerning the influence of martial arts on health and physical fitness, the psychological, social and moral factors involved, and the incidence of injury in these disciplines, were critically analyzed. Most of them showed a positive effect on health [34].

The authors point to the possibility of applying these exercises as a prescription in exercise therapy, also for the elderly. Martial arts as prescribed exercise may go from experience-based treatment to evidence-based treatment [1].

Our findings thus support the growing evidence that competitive level CS and MA are an effective means of improving people’s quality of life. Future research needs to clarify whether CS and MA can also be recommended to recreational and non-competitive practitioners as a means to improve their health and subjective quality of life.

### Limitations

There are several open questions and possible problems in the work which might contribute to further moderation of final conclusions. Considering net income per household in future research as well as determining the costs of participation in training, sports camps, trips to sports competitions or purchase of sports gear, would be helpful in more precise determination of how much the financial situation affects the quality of life of CS and MA practitioners. It would also be important to examine the reasons for practicing CS and MA, which would allow a broader explanation of the relationship between health behaviors and the quality of life of those who practice these disciplines

## 5. Conclusions

The combat sports and martial arts competitors were characterized by a slightly higher level of healthy behaviors than the individuals who practiced these disciplines recreationally or reconciled practicing various sports. They competitors were the least likely to smoke, half of them were nondrinkers, and most of them had dental checkups as recommended. The respondents had diverse assessments of the quality of life. The competitors (G III) were more satisfied with their health and evaluated their quality of life higher in the psychological, environmental and physical domains. In the physical domain, those who practiced sports recreationally (G I) and the competitors (GIII) had similar ratings. Those who practiced combat sports and martial arts along with other disciplines (G II) assessed their quality of life higher in the social domain. The people who displayed health-oriented behaviors had higher quality of life. The nonsmokers were more satisfied with their health. The nondrinkers and the nonsmokers rated their quality of life (social domain) lower. People who consumed alcoholic beverages rated their quality of life higher than the nondrinkers. In the social domain of the quality of life, the large effect size indicates differences between the nondrinkers and those who consumed high-alcohol and various beverages. The subjects who had had dental check-ups in the past 6–12 months or 1–2 years evaluated their quality of life (in the physical, social and environmental domains) higher than those who had visited the dentist earlier. The large effect size confirms these differences.

## Figures and Tables

**Table 1 ijerph-16-00875-t001:** Characteristics of people who practice combat sports and martial arts (independence χ^2^ test, Cramér’s V).

Variables	Combat Sports and Martial Arts (%)			Total (543)		*p* for χ^2^	Cramér’s V
	G I (*n* = 362)	G II (*n* = 115)	G III (*n* = 66)	*N*	%		
Age:						0.0220	0.1
<19	27.1	27.0	36.4	153	28.2		
20–23	19.9	31.3	19.7	121	22.3		
24–28	28.5	19.1	31.8	146	26.9		
>28	24.5	22.6	12.1	123	22.6		
Education:						0.0008	0.1
pre-secondary	16.6	11.3	28.1	90	16.8		
secondary	34.8	53.0	31.3	205	38.3		
post-secondary	48.6	35.7	40.6	240	44.9		
Financial situation:						0.0185	0.1
very good	35.5	24.6	39.4	181	33.7		
good	45.2	55.3	53.0	260	48.3		
fair	19.3	20.1	7.6	97	18.0		
Length of practicing sport:						0.0151	0.1
≤2 years	39.5	30.4	33.3	200	36.8		
>2 <4 years	23.8	15.7	13.6	113	20.8		
>4 <8 years	21.8	31.3	27.3	133	24.5		
≥8 years	14.9	22.6	25.8	97	17.9		
Weekly exercise time:						0.0000	0.2
≤180 min	26.8	20.8	3.0	123	22.7		
181–360 min	42.5	32.2	27.3	209	38.5		
361–540 min	14.9	17.4	27.3	92	16.9		
541–720 min	7.5	14.8	16.7	55	10.1		
≥721 min	8.3	14.8	25.7	64	11.8		
Weekly exercise frequency:						0.0000	0.2
twice a day	4.4	7.0	13.6	33	6.1		
once a day	16.0	24.3	42.4	114	21.0		
3–4 times a week	51.7	46.1	39.4	266	49.0		
1–2 times a week	27.9	22.6	4.6	130	23.9		

**Table 2 ijerph-16-00875-t002:** Selected healthy behaviors of people who practice combat sports and martial arts (independence χ^2^ test Cramér’s V).

Variables	Combat Sports and Martial Arts (%)			Total		*p* for χ^2^	Cramér’s V
	G I (*n* = 362)	G II (*n* = 115)	G III (*n* = 66)	*N*	%		
Smoking (ever)						0.0224	0.1
smokers	24.6	30.3	11.5	126	24.2		
nonsmokers	75.4	69.7	88.5	394	75.8		
Consumption of alcoholic beverages						0.0174	0.1
nondrinkers	41.0	27.2	50.0	212	39.2		
low-alcohol	38.5	46.5	33.3	214	39.6		
high-alcohol	18.0	18.4	13.6	95	17.6		
various	2.5	7.9	3.1	20	3.7		
BMI (kg/m^2^)						n.s.	0.1
<20.0	11.0	14.0	16.4	67	12.3		
[20.0; 25.0)	52.2	62.3	56.7	298	54.9		
[25.0; 30.0)	31.5	21.9	20.9	153	28.2		
≥30	5.3	1.8	6.0	25	4.6		
Dental checkups						0.0012	0.1
in the last 6 months	50.7	59.7	66.8	295	54.5		
6–12 months ago	29.6	16.7	12.1	134	24.8		
1–2 years ago	17.7	16.7	19.7	96	17.7		
earlier	1.9	7.0	1.5	16	3.0		

**Table 3 ijerph-16-00875-t003:** Satisfaction with life, health and particular domains of the quality of life (WHOQOL-BREF) of people who practice combat sports and martial arts (test H, $E_{R}^{2}$, test U, rg).

Specification	Groups	G II	G III	G II	G III	Rank Means
		Value of *p* for U Statistics		Glass Rank Biserial Correlation (rg)		
Satisfaction with life H(2.543) = 2.314 $E_{R}^{2}$ = 0.0043 *p* = 0.3143	I	0.3650	0.4133	0.1	−0.1	273.13
	II		0.1696		−0.4	257.79
	III					290.53
Satisfaction with health H(2.543) = 5.321 $E_{R}^{2}$ = 0.0098 *p* = 0.0699^t^	I	0.1478	0.2131	0.1	−0.1	273.97
	II		0.0403 *		−0.5	249.74
	III					299.98
Physical domain H(2.543) = 12.828 $E_{R}^{2}$ = 0.0237 *p* = 0.0016	I	0.0008 *	0.6175	0.2	−0.1	282.61
	II		0.0046 *		−0.7	226.22
	III					293.55
Psychological domain H(2.542) = 7.488 $E_{R}^{2}$ = 0.0138 *p* = 0.0237	I	0.8688	0.0761	0.1	−0.2	273.10
	II		0.0074 *		−0.7	244.47
	III					309.87
Social domain H(2.543) = 4.240 $E_{R}^{2}$ = 0.0078 *p* = 0.1200	I	0.0487 *	0.3320	−0.1	−0.1	262.51
	II		0.7704		0.1	294.94
	III					284.08
Environmental domain H(2.543) = 15.091 $E_{R}^{2}$ = 0.0278 *p* = 0.0005	I	0.0092 *	0.0238 *	0.2	−0.2	275.49
	II		0.0001 *		−1.0	231.45
	III					323.52

* statistically significant for *p* ≤ 0.05.

**Table 4 ijerph-16-00875-t004:** Satisfaction with health and the quality of life (social domain) (WHOQOL-BREF) of people who practice combat sports and martial arts, depending on smoking (test H, $E_{R}^{2}$, test U, rg).

Specification	Group	Values of *p* for U Statistics		Glass Rank Biserial Correlation (rg)		Rank Means
	Smoking	Smokers	Nonsmokers	Smokers	Nonsmokers	
Satisfaction with health H(1, 520) = 6.959; $E_{R}^{2}$ = 0.0134 *p* = 0.0083	Smokers		0.0155 *		−0.1	232.3
	Nonsmokers					269.51
Social domain H (1, 520) = 3.974; $E_{R}^{2}$ = 0.0077 *p* = 0.0462	Smokers		0.0480 *		0.1	283.54
	Nonsmokers					253.13

* statistically significant for *p* ≤ 0.05.

**Table 5 ijerph-16-00875-t005:** The quality of life (WHOQOL-BREF) of people who practice combat sports and martial arts depending on the consumption of alcoholic beverages (test H, $E_{R}^{2}$, test U, rg).

Specification	Consumption of Alcoholic Beverages	Values of *p* for U Statistics			Glass Rank Biserial Correlation (rg)			Rank Means
		Low-Alcohol	High-Alcohol	Various Beverages	Low-Alcohol	High-Alcohol	Various Beverages	
Social domain H(3, 539) = 34.282; $E_{R}^{2}$ = 0.0066 *p* = 0.0000	Nondrinkers	0.0000 *	0.0000 *	0.3492	−0.3	−0.6	−0.6	222.83
	Low-alcohol		1.0000	1.0000		−0.2	−0.0	293.14
	High-alcohol			1.0000			0.4	318.07
	Various							293.47

* statistically significant for *p* ≤ 0.05.

**Table 6 ijerph-16-00875-t006:** The quality of life (WHOQOL-BREF) of people who practice combat sports and martial arts depending on having dental checkups (test H, $E_{R}^{2}$, test U, rg).

Specification	Dental Checkups	Values of *p* for U Statistics			Glass rank Biserial Correlation (rg)			Rank Means
		>6<12 Months	>1<2 Years	Earlier	>6<12 Months	>1<2 Years	Earlier	
Physical domain H(3, 541) = 13.974; $E_{R}^{2}$ = 0.0259 *p* = 0.0029	Past 6 months	1.0000	0.0019 *	1.0000	0.01	0.3	0.2	284.14
	>6 <12 months		0.0142 *	1.0000		0.6	0.3	281.54
	>1 <2 years			1.0000			−0.7	218.03
	Earlier							258.25
Social domain H(3, 541) = 11.676; $E_{R}^{2}$ = 0.0216 *p* = 0.0086	Past 6 months	0.0070 *	0.7724	1.0000	0.2	0.1	−0.1	288.71
	>6 <12 months		1.0000	0.7370		−0.2	−0.7	235.89
	>1 <2 years			1.0000			−0.7	260.81
	Earlier							299.69
Environmental domain H(3, 541) = 9.817; $E_{R}^{2}$ = 0.0182 *p* = 0.0202	Past 6 months	0.0078 *	0.0231 *	0.8271	0.2	0.2	0.1	289.39
	>6 <12 months		0.9487	0.3806		−0.0	−0.5	246.55
	>1 <2 years			0.3758			−0.6	246.8
	Earlier							281.91

* statistically significant for *p* ≤ 0.05.

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
