# Peer review of "Selected Healthy Behaviors and Quality of Life in People Who Practice Combat Sports and Martial Arts"

_ijerph, 2019, doi:10.3390/ijerph16050875_

Reviewer 1 Report

I read your manuscript with a great deal of interest and believe it fits the special issue really well. I believe you must redo your manuscript to include effect size values. All papers need effect size values. It seems impact factor journals really need effect size values to maintain high publishing standards.

You need in a section in your methods to detail right around lines 124-131 to indicate your effect size plan. You have the cascade of logical options - partial eta squared via SPSS for your F tests, and then Hedges' g is the logical choice for between or within a group effect size calculation so G I vs. G II and so on.

This will take some time given the volume of tests and comparisons. I think it is a must so the reader as well of yourself can gauge meaningfulness of the significant differences.

Author Response

Dear Editor,

Thank you very much for proposed corrections. We have substantially revised the manuscript according to your suggestions.  Below are the detailed answers to issues which were raised.

Reviewer 2 Report

The paper presents a study that analysed the relationship between health behaviors and the quality of life of people who practice combat sports and martial arts. This has a roughly sample. However, the introduction and discussion are not persuasive enough that the findings make a significant contribution to the literature and could therefore override these limitations. I include some comments below related to this summary for consideration.

1. In relation to the contribution of the study to the literature, I did not get a sense from the article that the findings revealed anything other than what we already know. Please clarified that;

2. The introduction of the paper was very descriptive, it did not situate the current study in literature or highlight what the gap in the literature is that this study is trying to address. At least, the authors should situate better the main purpose of this study;

3. The statistical analysis is inconclusive. If the main objective of the present study was to analyse the relationship between quality of life who people, what is the rationale for using the comparison tests?

4. The discussion is very descriptive and any statements about the contribution and conclusions of the study are not new. At least this moment. Please clarified better and justified your choices.

Author Response

(The authors gave the same response as above.)

Reviewer 3 Report

Dear authors,

congratulations on an overall well done study. I've enjoyed reading it very much. There were a few possible minor spelling mistakes that I will list at the end, after noting two slightly larger points that I found problematic.

(1) The first problematic aspect was my feeling that the formulation of the final conclusion as noted at the very end of the abstract and the discussion section are formulated in a bit too strong manner, and in this way not directly related to your study's findings. It seems to me that you have done a correlational study, but to argue that your findings 'require that these disciplines be promoted' (Abstract, p.1, line 30 & Discussion, p. 9-10, line 342-343) is a claim that is too strong, and based on causative assumptions for which you have not tested in your study and through its design. My recommendation would be to express this in a more moderate fashion, e.g. by stating something like the following, though you might want to reformulate this in your own way: 

'We have found positive correlations between combat sports and martial arts, health behaviours and higher scores ins subjective quality of life measures, particularly where practitioners are exclusively focussed on CS and MA and practice these at a competitive level. Our findings thus support the growing evidence for competitive CS and MA as an effective means for improving people's QoL. Future research needs to clarify whether CS and MA can also be recommended to recreational and non-competitive practitioners as a means to improve their subjective quality of life.’

Similarly, in the discussion section, I would rather say, for example, ‘associations between the healthy behaviours and the QoL of CS and MA practitioners suggest that their practice, particularly at high, or competitive levels, could contribute to the health of society.’ But here as well, I would recommend some nuanced commentary on recommendations for future research.

Finally, I think this issue regarding your conclusions should also be reflected in the formulations of your hypothesis 1 & 2. I would recommend not using the term 'athletes' because it is not well defined in your study and too broad for it. You might simply say e.g. (1) 'people' or 'CS/MA practitioners ... who practice at a competitive level'. Hypothesis 2 needs to clarify who CS/MA practitioners have higher QoL than, and then you have to clarify if this was actually tested in your study.

(2) Reading your article, I would have also welcomed a more critical engagement with questions regarding the relationship and relative importance of CS and MA training versus occupational status and net income per household. As you yourselves mention, the latter two points are particularly important factors to people’s QoL. If I have understood correctly, 48.3% of your cohort reported a good financial situation and GIII had some of the highest % in this regard. Questions of interest (maybe for future research?) then would be, for example: How much of these people’s QoL can be attributed to their financial situation vs their CS/MA practice? Also, is it necessary to have a good to very good financial situation in order to practice CS/MA at a competitive level (maybe to be able to pay for practice, competition expenses, or have the available free time)? And if this were the case, does it make sense to recommend these practices as a means to improve QoL, or is the more fundamental method that of improving people’s financial situations, which in turn enable them to make further improvements via the competitive practice of CS/MA? These are just a few examples for questions in this direction, in which there are also many more. Unless I have misunderstood your findings, I think it would improve the quality of your study if you acknowledged that there are a range of open questions and possible problems here, and this might also contribute to further moderating your final conclusions. 

The broader point here, beyond questions about finances and QoL, is probably that it would be good if you wrote at least a small section on the limitations of your study and directions for future research that could be done to build on its findings.

(3) There are a range of other fairly strong claims throughout your article that I would consider making a little less strong:

Page 1, line 13: which could help reduce aggression

Page 7, line 222: Is choosing healthy behaviours really 'essential' to QoL given what you have found with regard to alcohol consumption. Maybe rather 'choosing healthy behaviours contributes to ...'?

Page 7, line 233: Maybe a larger point affecting your study more broadly speaking, but I am not sure that you can say that boxing and MMA are based on 'traditional philosophy'? If so, which philosophy is this, and is it really the same as in the other disciplines you have included? Can we truly speak of CS/MA as if they were all one and the same phenomenon? I would simplify this point by simply writing sth like" 'These kinds of sports require willpower, discipline, and perseverance in action'.

(4) Possible grammar and spelling mistakes

Page 1, line 16: Were your participants, people who practice, or practised CS and MA? This might be a significant difference.

Page 2, line 51: instead of ‘reveal the’, maybe write ‘have shown’?

Page 2, line 63-64: healthy populations

Page 8, line 288: In a study of teenagers…

I hope this helps and wish you the best on your continuing journey in martial arts and health research.

Author Response

Dear Editor,

Thank you very much for proposed corrections. We have substantially revised the manuscript according to your suggestions.  Below are the detailed answers to issues which were raised.

Round  2

Reviewer 1 Report

Thank you for your efforts in the revision process. I enjoyed reading your revised manuscript.

Author Response

Thanks.

Reviewer 2 Report

The paper should be accept.

Author Response

Thanks.